# Toward Smart, Automated Junctional Tourniquets—AI Models to Interpret Vessel Occlusion at Physiological Pressure Points

**DOI:** 10.3390/bioengineering11020109

**Published:** 2024-01-24

**Authors:** Guy Avital, Sofia I. Hernandez Torres, Zechariah J. Knowlton, Carlos Bedolla, Jose Salinas, Eric J. Snider

**Affiliations:** 1U.S. Army Institute of Surgical Research, JBSA Fort Sam Houston, San Antonio, TX 78234, USA; 2Israel Defense Forces Medical Corps, Ramat Gan 52620, Israel; 3Division of Anesthesia, Intensive Care, and Pain Management, Tel-Aviv Medical Center, Affiliated with the Faculty of Medicine, Tel Aviv University, Tel Aviv 64239, Israel

**Keywords:** machine learning, ultrasound imaging, artificial intelligence, tourniquet application, hemodynamics, tissue phantom, hemorrhage

## Abstract

Hemorrhage is the leading cause of preventable death in both civilian and military medicine. Junctional hemorrhages are especially difficult to manage since traditional tourniquet placement is often not possible. Ultrasound can be used to visualize and guide the caretaker to apply pressure at physiological pressure points to stop hemorrhage. However, this process is technically challenging, requiring the vessel to be properly positioned over rigid boney surfaces and applying sufficient pressure to maintain proper occlusion. As a first step toward automating this life-saving intervention, we demonstrate an artificial intelligence algorithm that classifies a vessel as patent or occluded, which can guide a user to apply the appropriate pressure required to stop flow. Neural network models were trained using images captured from a custom tissue-mimicking phantom and an ex vivo swine model of the inguinal region, as pressure was applied using an ultrasound probe with and without color Doppler overlays. Using these images, we developed an image classification algorithm suitable for the determination of patency or occlusion in an ultrasound image containing color Doppler overlay. Separate AI models for both test platforms were able to accurately detect occlusion status in test-image sets to more than 93% accuracy. In conclusion, this methodology can be utilized for guiding and monitoring proper vessel occlusion, which, when combined with automated actuation and other AI models, can allow for automated junctional tourniquet application.

## 1. Introduction

Hemorrhage is the leading cause of preventable death in trauma casualties, both civilian [1] and in combat [2]. Through public health programs such as the “Stop the Bleed” initiative [3] and the abundant use of tourniquets, the mortality from extremity hemorrhage has greatly diminished [4]. However, junctional hemorrhage remains a largely unsolved problem. Junctional hemorrhage is defined as hemorrhage from the areas connecting the extremities to the torso—axillae, shoulders, groin, buttocks, and proximal thighs, as well as from the neck [5]. As these areas are not amenable to traditional extremity tourniquet placement, several alternative solutions are in use.

The first approach is wound packing, which creates pressure inside the bleeding wound in order to press against the bleeding vessel, with materials ranging from gauze supplemented with pro-coagulant substances [6] to expanding sponges inserted by a syringe into the wound [7]. While these techniques are effective against hemorrhage from a venous source, they lack efficacy against hemorrhage from a major artery, such as the femoral artery [8]. Another approach is manual pressure points wherein a major artery, proximal to the hemorrhage source, is pressed against a bony surface to stop the blood flow to the wound and beyond. While a single study [9] has described this practice as ineffective, leading to its elimination from most clinical practice guidelines, more recent studies show promising results [10,11]. However, this technique requires continuous monitoring and pressure by the provider, which may limit its effectiveness when used in prolonged case situations. Therefore, this approach provides only a short temporizing solution and severely limits the ability to transport and evacuate the patient.

Several junctional tourniquets have been developed utilizing the pressure points concept. These devices are designed to maintain ongoing pressure on the vessel, pending a definitive surgical solution. Currently, there are four junctional tourniquets that are approved by the Food and Drug Administration: abdominal aortic junctional tourniquet (AAJT), junctional emergency treatment tool (JETT), SAM^®^ junctional tourniquet, and Combat Ready Clamp (CRoC) [12,13,14]. The AAJT consists of a windlass mechanism to stabilize the device on the abdomen, axilla, or groin, where then a pneumatic bladder is inflated to occlude the artery. The CRoC is a compression clamp that can be placed over the axilla or groin, where it is then tightened by a hand crank. The JETT is a belt that can be placed over the pelvis, where two pads can be mechanically extended by turning handles. The SAM junctional tourniquet consists of inflatable bladders that are placed over the pelvis with a belt. Each of these tourniquets is recommended to be used for less than four hours for axilla and groin placements, while less than one hour is recommended for the abdomen [5,12]. However, studies have shown their utilization to be cumbersome and time consuming [15,16] and they have a high failure rate in training [17] and real combat scenarios [18]. Moreover, their effectiveness drops dramatically during patient transport [19].

A few reports on the use of ultrasound to guide pressure against an artery have been published. Garrick et al. have reported a 93% initial success rate in occluding the femoral artery using an ultrasound probe [20]. A case report has described the use of an ultrasound probe against the abdominal aorta to mitigate an iliac artery hemorrhage [21]. However, sonography requires skill, precluding it from prevalent use, compared to an extremity or a junctional tourniquet.

We hypothesized that the skill threshold could be overcome through the use of artificial intelligence (AI) models that can detect the appropriate vessel requiring occlusion from real-time ultrasound video feeds. AI has begun to revolutionize medicine through smart, precision-medicine applications such as categorizing a wide range of abnormal states from optical coherence tomography scans [22], compiling large diverse data sets for making medical decisions [23], and using predictive text AI models to provide medical recommendations during telemedicine [24]. Smart medicine applications have been extensively reviewed elsewhere [25,26,27,28,29,30]. Focusing on AI for interpreting ultrasound images [31], applications include the identification of tumors [32], diagnosing infectious disease [33,34], and determining eFAST scan outcomes [35,36], among others. Each of these applications often relies on deep convolutional neural networks, which extract image features and parameter weights to identify differences in images. Alternatively, object detection AI models can produce a bounding box overlay [37] around features for tracking regions of interest in an ultrasound image, such as tumors [38] and vessels [39]. Each of these approaches may be applicable to this use case as AI models can potentially be trained to guide a user to the proper pressure point and ensure that enough force is applied for occlusion. In this research effort, we describe a deep-learning image classification algorithm to analyze sonographic images and provide the occlusion status of a vessel of interest. This can be used to aid the user in pressing the artery until occlusion and monitoring the effectiveness of this pressure. This algorithm is meant to be integrated into an ultrasound probe, making this probe an effective pressure head as part of a junctional tourniquet, without the need of expertise. The constant monitoring of effectiveness will allow for a rapid or automated response to displacement, overcoming this issue when transporting the patient. This work is presented by first describing the identification of the occlusion threshold in the phantom and capturing images for AI training. Then, we compare the performance metrics of two separate AI model architectures. Lastly, we validate the best performing model using an ex vivo swine model.

## 2. Materials and Methods

### 2.1. Tissue Phantom Setup

An ultrasonically compatible basic phantom was developed for image collection to train and test a classification algorithm. This phantom was created with 10% clear ballistic gel (CBG) (Clear Ballistics, Greenville, SC, USA) using a 3D printed mold. Due to the high temperature needed to melt the CBG, a co-polyester filament with a fused deposition 3D printer (Ultimaker, Utrecht, The Netherlands) or high-temp resin and stereolithography 3D printer (FormLabs, Somerville, MA, USA) were selected to print the mold. The phantom was fashioned as a 3″ length × 3″ width × 2.5″ depth box around a 3″ length × 2″ wide × 1” depth wax block (McMaster-Carr, Elmhurst, IL, USA), which acted as a bone to allow for vessel occlusion.

The CBG was then cut into small pieces, placed into a 500 mL beaker, and melted at 130 °C using a laboratory oven (Thermo Fisher Scientific, Waltham, MA, USA) for approximately 2 h or until the gel was fully melted and de-bubbled. Using a ¼” OD biopsy punch (McMaster-Carr, Elmhurst, IL, USA) to hold the place of a vessel, the CBG was slowly poured into the mold lined with silicone oil (Sigma-Aldrich, St Louis, MO, USA) and left to cool at room temperature. Once cooled, the phantom was removed from the mold and placed over the wax block.

### 2.2. Tissue Phantom Imaging

The phantom was fitted with a ¼” diameter thin-walled latex tubing (GF Health Products, Atlanta, GA) to act as a vessel, which was connected to a simple flow loop. This loop consisted of a simple peristaltic pump (Masterflex, Gelsenkirchen, Germany) driving Doppler-compliant fluid (CIRS Tissue Simulation Technology, Norfolk, VA, USA) from a reservoir through the phantom and a pressure sensor (ADInstruments, Sydney, NSW, Australia) that connected directly to a data acquisition unit (ADInstruments, Sydney, NSW, Australia). Between the pump and the phantom there was a bypass line so that flow could be diverted during occlusion while maintaining physiological relevant pressure in the system. The phantom was kept underwater for imaging. Ultrasound imaging was performed using a 15L4A probe (Terason, Burlington, MA, USA) from a Terason 3200T Plus US imaging system (Terason, Burlington, MA, USA). Real-time video feed from the US screen was recorded with the LabChart (ADInstruments, Sydney, NSW, Australia) software using a video capture box (Amazon, Seattle, WA, USA). US images were recorded with and without color Doppler overlay. The Doppler modality helped to confirm the presence or absence of flow in the vessel. Distal pressure to the phantom was captured in real time to determine the pressure reduction achieved by tourniquet application, which is analogous to flow reduction. With the vessel in view and the phantom placed on top of the wax block, the ultrasound probe was used to compress the vessel until flow was stopped or reduced to at least 90% of its initial rate.

### 2.3. Ex Vivo Swine Model Setup

Euthanized swine tissue was procured from a commercial vendor (Animal Technologies, Tyler, TX, USA) from the lumbar area to the knees as this section allowed enough area for experimental setup. Swine tissue was utilized due to the similarities between human and swine femoral vessels [40,41,42]. Using previously developed methodologies [43], we cannulated the distal superficial femoral artery using a 14G catheter (MedOfficeDirect, Naples, FL, USA). Proximal cannulations for both vessels were made using 8Fr PCI introducers (Argon Medical Devices, Athens, TX, USA) and secured using silk ligatures. An arteriovenous shunt was fashioned from tubing and a 3-way stopcock was used to connect the vessels distally and allow flow going through the artery to return through the vein or be hemorrhaged from the system. The hemorrhage site was connected to a pressure sensor for measuring distal pressure to the occlusion site. The proximal catheters were connected to a heart-mimicking pump (SuperPump AR Series, Vivitro Labs, Victoria, Canada) to create a flow loop. Doppler-compliant fluid was pumped through the loop to enable Doppler modality for ultrasound imaging. A Draeger patient monitor (Delta XL, Lubeck, Germany) was used to monitor the pressure in the flow loop so that flowrate adjustments to the pump could alter pressure in the system to achieve a systolic and diastolic pressure of approximately 110 and 80 mmHg, respectively. A pressure sensor (ICU Medical, San Clemente, CA, USA) connected to a data acquisition unit was connected downstream in the model, between the artery and arteriovenous shunt. Similar to the phantom model, a Terason or Sonosite Edge (Fujifilm Sonosite, Bothell, WA, USA) US system was used to capture live ultrasound feed using LabChart software and a capture box. Once the vessels were in view, pressure was applied on the inguinal crease, compressing the vessels until 90% occlusion was achieved.

### 2.4. Ultrasound Image Processing

After data were collected, videos were exported from LabChart as MP4 files and mean distal pressure readings were downsampled to 10 Hz to match the frame rate for the recorded video. Using MATLAB (v2022a, Mathworks, Natick, MA, USA), mean pressure vs. time data were plotted for each recording, and three regions were identified—(i) start and (ii) end region for unobstructed pressure measurement, and (iii) end of probe occlusion of the vessel. A mean unobstructed pressure was measured for the full-flow region, which was then used to create gates for the classification categories. For a two-class scenario, full-flow and no-flow categories were separated by a percent reduction in the distal pressure, ranging from 90 to 50% threshold in different experimental setups. This was used to determine the best threshold to distinguish between full-flow and no-flow classes. For the three-class scenarios, full flow was characterized as unobstructed flow to only 10% reduction in mean pressure, partial flow was 10% reduction to the determined no-flow marker (50 to 90% reduction, depending on the experimental setup), and no flow was any pressure reduction below this threshold. During data processing, images were cropped to remove ultrasound user interface information from the image and then resized to 512 × 512 × 3. This process was repeated for each recorded ultrasound video for tissue phantom and swine.

### 2.5. Neural Network Model Training

All neural network model developments and evaluations were performed using MATLAB v2022a on an AMD Ryzen 9 5900HX 3.3GHz, 32 GB RAM, and NVIDIA RTX 3080 16 GB VRAM computer system (Lenovo, Morrisville, NC). Two neural network architectures were used: (1) a previously developed custom classification network, ShrapML, optimized for ultrasound image interpretation [44,45]; and (2) MobileNetV2, a conventional neural network model that performed best for interpretation of ultrasound images for small datasets [46]. ShrapML was Bayesian-optimized for ultrasound applications. The optimized architecture comprised 6 blocks each containing a convolutional neural network layer, rectified linear unit activator, and max pooling layer, followed by a flattening layer, fully connected layer, and 36% dropout layer that led to a classification output setup [45]. During the optimization process, the Root Mean Squared Propagation (RMSprop) optimizer, which is a widely used optimizer for classification tasks, was selected as optimal. Each model was fitted with a 512 × 512 × 3 image input layer and a two- or three-category classification output layer, depending on the image sets used.

For phantom training, image sets were loaded and randomly split 80:20 for training and validation, while a phantom image set was completely held out for blind testing. For swine training, a single image set was loaded and randomly split 60:20:20 for training, validation, and testing, respectively. In some training cases, data augmentation in the form of affine transformations was randomly introduced to training images. Specifically, reflections and translation in the X- (−128 to 128 pixels) or Y-direction (−64 to 64 pixels) were introduced randomly in these data augmentation training scenarios.

Model training was performed in all instances for up to 100 epochs using a RMSProp optimizer with a 0.001 learn rate. A batch size of 32 was used throughout with evaluation of validation loss performed at the end of each epoch. A validation patience of five was used, which meant if the validation loss was not further reduced in five epochs, training ended early, and the lowest validation loss was selected as the optimal model. Training was repeated three times with different random image splits for each training strategy, and each model was independently evaluated for determining overall performance.

### 2.6. Evaluation of Neural Network Model Performance

Model performance was evaluated with test images held out from the training and validation process. Predictions and confidences were calculated for each test image and compared to ground truth labels in order to build a confusion matrix using GraphPad Prism 9 (San Diego, CA, USA). For two-category models, positive predictions were no-flow or occlusion images, while negative predictions were full-flow images. Using these identifications, accuracy, precision, recall, specificity, and F1 scores were calculated. Confidences were used to construct a receiver operating characteristic (ROC) curve and to measure the area under the ROC (AUROC). Performance metrics were found for triplicate models and were shown as average values throughout.

In addition, Gradient-weighted Class Activation Mapping (GradCAM) overlays were created for 1/24th of the testing images for each model. GradCAM overlays are used to produce an approximate localization heat map identifying “hot spots” for regions important to the model prediction as a means of making models more explainable and confirming irrelevant image artifacts are not being tracked [47]. This technique is widely used for showing how AI models used for medical image interpretation are tracking the same area of interest as an expert would [48,49,50]. For our study, GradCAM overlays were created using a built-in MATLAB command for 1/24th of the test images and were saved according to the ground truth and prediction labels. Representative images were selected for highlighting regions of the images the models identified when making a classification prediction.

## 3. Results

### 3.1. Determination of the Optimal Threshold for Occlusion

To develop a machine learning model for monitoring junctional flow occlusion, we first identified the occlusion threshold most suitable for distinguishing flow (negative) and no-flow (positive) conditions. Using a tissue phantom model, training performance was compared with thresholds set at 50, 60, 70, 80, or 90% distal pressure reduction for occlusion (Table 1). Lower threshold values had higher accuracy and improved performance in most performance metrics, while the 80 and 90% threshold conditions’ performances were reduced. As the highest occlusion threshold is ideal, 70% was selected as the threshold for future testing as the differences were minimal with lower thresholds but this avoided the performance reduction in the higher thresholds.

### 3.2. ShrapML and MobileNetV2 Performance for Tracking Tissue Phantom Vessel Occlusion

Next, various deep learning model setups were used for classifying junctional ultrasound images for flow or no flow, following tourniquet application (Figure 1). Two different model architectures were used—ShrapML and MobileNetV2—each without and with affine transformations for data augmentation. MobileNetV2 models trended toward no-flow (false positive) predictions, resulting in recall metrics of 0.675 and 0.646 without and with data augmentation, respectively (Table 2). However, MobileNetV2 was strong at identifying full-flow conditions, with specificity reaching 0.990 and 0.996 without and with data augmentation, respectively. In contrast, augmentation had a more pronounced effect on ShrapML training. Without augmentation, ShrapML models had a high false negative (full-flow prediction) rate, with a specificity of 0.683. Utilizing data augmentation for ShrapML training solved this false negative bias, increasing specificity to 0.991 without impacting the false positive rate. Overall, ShrapML models with augmentation had the strongest accuracy (0.934) and F1 score (0.918) metrics and were selected as the optimal configuration for this application.

To further understand model performance, we created GradCAM overlays to highlight the regions of the ultrasound images most critical to the model prediction (Figure 2). When looking at full-flow ultrasound images, most of the models accurately tracked the vessel patency as the key feature, except for the ShrapML model without augmentation for the training data. The extent of tracking the vessel was reduced in images where less or no Doppler signal was present. For the no-flow image class, model trends were less consistent. MobileNetV2 models were more frequently tracking features at the edges of the image (no augmentation) or below the tissue phantom (with augmentation). ShrapML without augmentation had no strong feature correlation, indicating no flow was identified as an absence of key feature extraction. ShrapML with data augmentation successfully tracked the compression of the tissue phantom, but the precise feature being tracked was not obvious.

### 3.3. Effect of Three Classes on ShrapML Model Performance for Tracking Junctional Vessel Occlusion

An alternative model design was assessed, which added a third category as partial flow, which represented 10% to 70% distal pressure reduction. This improved the full-flow and no-flow true prediction rate (Figure 3). However, this was at the expense of the partial-flow category as over 75% of the predictions were incorrect. This was further highlighted through GradCAM overlays. The full-flow and no-flow identified images were still tracking the vessel placement and phantom compression, respectively (Figure 3). The partial-flow class did not identify any obvious trends in the ultrasound image, including frequently tracking features outside of the tissue phantom. As a result, the two-category methodology was identified as most suitable for this application.

### 3.4. Performance of ShrapML Model for Tracking Junctional Vessel Occlusion in an Ex Vivo Swine Model

Lastly, the ShrapML binary classification network design was retrained for use with junctional vessel occlusion datasets collected from the ex vivo swine model. Multiple ultrasound clips were collected from a single ex vivo swine subject, and 20% of the images were held out for testing model performance. Generally, the models had a similar performance to the tissue phantom, with a slight bias toward false positive, no-flow predictions (Figure 4). Results across the three replicate trained models were consistent, each with a similar AUROC (Figure 4) and with low standard deviations for each performance metric (Table 3). Accuracy was over 90% for swine image sets, similar to the tissue phantom performance. For comparison, the models were also trained using the more aggressive 90% distal pressure reduction threshold for occlusion. Overall, performance was minimally impacted when using swine images with this higher occlusion threshold (Figure 4B,C). GradCAM overlays were used to track if the features identified by the model were tracking the vessel as it occluded (as observed in the tissue phantom) or overfitting to noise in the images (Figure 5). In full-flow images, the artery and vein were sometimes being tracked, while at other times the artery was the primary feature responsible for model predictions. In the no-flow image class, the trends were less obvious, but in general the bottom tissue features were tracked if they appeared higher in the ultrasound image due to tissue compression. This provided proof of concept that the model can work in animal tissue, but additional images and subject variability will be needed for more robust performance.

## 4. Discussion

Junctional and pelvic hemorrhage continue to be a significant cause of early preventable death among trauma casualties, and there is a need for a solution that can provide rapid and reliable hemostasis without requiring advance expertise. Here, we have demonstrated the first part of the development of an artificial intelligence algorithm that can serve to guide such a device, and that can confirm appropriate pressure and continuously monitor the effectiveness of pressure.

An important step in setting up the training datasets for this application was defining a threshold for flow or distal pressure decrement. We approached this from an AI training perspective and a threshold of 70% resulted as the most optimal for model performance; however, the reduction in performance may be preferred if a higher 90% occlusion threshold could be tracked. Both were evaluated in the swine image sets at more than 90% accuracy. More tuning is needed to settle on the optimal threshold. The threshold decision may also impact the output category design of the model. We evaluated two-category (full flow and no flow) and three-category (additional partial-flow category) classifier models, and the partial-flow classification had poor prediction performance. However, this is likely due to the significantly low quantity of images in that partial occlusion window. More data may help with training this intermittent category, which may be preferred in a junctional tourniquet design so that progress toward the goal can be tracked. This may also be accomplished by the use of a regression deep-learning model output as an output to categorical outputs. This may allow for tracking the percent occlusion as opposed to arbitrary categories defining proper occlusion.

The use of AI for ultrasound guidance of medical interventions is not unique—it has been used, for instance, to guide central vascular access. However, to our best knowledge, its use for hemorrhage control has never been described before, potentially due to the relatively high cost of ultrasound machines. With the gradual decrease in its cost and size, the use of ultrasound for hemorrhage control begins to appear feasible, making expertise the limiting factor for use. AI offers a pathway to overcome this limit, allowing healthcare providers, or even laypersons, not trained in sonography to utilize ultrasound technology for this application. The AI models in this work were successfully developed for tracking occlusion, a critical first step in automating a junctional tourniquet.

Another significant advantage of AI over “standard” use of ultrasound for this purpose is its ability for continuous monitoring—the ultrasound system maintains “visualization” of the obstructed vessel and can raise an alarm if this obstruction is no longer effective. In the absence of such an alarm, the first sign of failure might be a pool of blood forming under the casualty, or clinical deterioration in the casualty’s mental status or vital signs, all signifying loss of a substantial amount of precious blood.

This work has several limitations. First, in this work we show an algorithm that can identify appropriate pressure on the artery, but not guide the user to position the probe in the right location. This work is already in progress and will be discussed in future studies. Second, it is based on a simple phantom model, without complex anatomical features. This approach of initial training using phantom data has been shown to be effective in reducing the requirement of animal or human data [51], and the performance demonstrated in this study on the ex vivo model supports that. However, acquisition of human data will be necessary for further development. Lastly, the AI models developed currently rely on color Doppler overlays, which may not be available as a feature on smaller, more portable ultrasound devices. Prototypes will need to ensure that this feature is present, or AI models can potentially be trained to bypass the need for this dependency.

## 5. Conclusions

Controlling junctional hemorrhage beyond current technology is a critical need for trauma care for both military and civilian situations. The technology we have demonstrated in this work highlights how an AI algorithm for monitoring vessel occlusion has the potential to improve junctional tourniquet application. AI models were developed to track vessel occlusion in phantom and ex vivo swine occlusion to more than 90% accuracy. Inclusion of this AI model with an engineered prototype for actuating compression will allow for automated vessel compression and real-time monitoring of tourniquet efficacy. Future studies will evaluate effectiveness on animal models and/or human volunteers. Further advancement of this technology will simplify junctional tourniquet use and help reduce the high mortality associated with junctional hemorrhage.

## 6. Patents

G.A and E.J.S. are inventors on a filed provisional patent owned by the U.S. Army related to the automated junctional tourniquet (filed 27 October 2023).

## Figures and Tables

**Figure 1 bioengineering-11-00109-f001:**
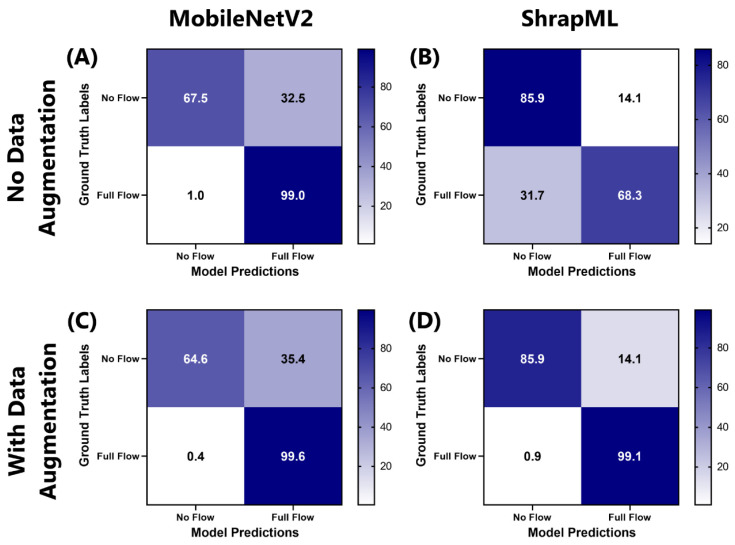
Confusion matrices for MobileNetV2 and ShrapML models tracking junctional vessel occlusion. Average confusion matrices (n = 3 trained models) are shown for (**A**,**C**) MobileNetV2 and (**B**,**D**) ShrapML models, (**A**,**B**) without data augmentation and (**C**,**D**) with data augmentation. Confusion matrix values are expressed as percentages across each ground truth category.

**Figure 2 bioengineering-11-00109-f002:**
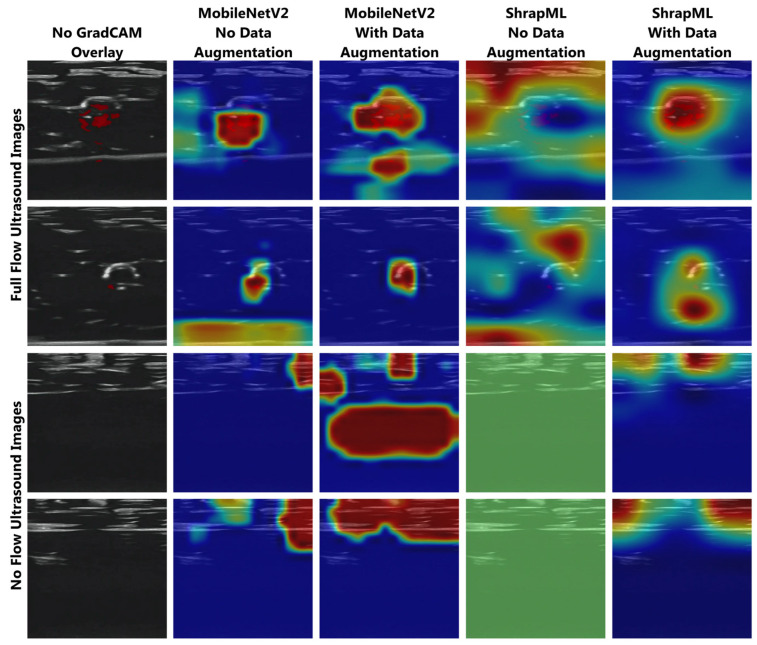
Gradient-weighted class activation maps (GradCAM) for trained binary classifier models for ultrasound tracking of junctional vessel occlusion. (Column 1) Base ultrasound images are shown for reference as well as (left to right) GradCAMs for Mobile Net V2 without and with data augmentation, and ShrapML without and with data augmentation. Representative ultrasound images are shown for full-flow and no-flow categories. Areas with high relevance to model predictions are highlighted by red-yellow overlays, while lower relevance regions are highlighted in blue-green.

**Figure 3 bioengineering-11-00109-f003:**
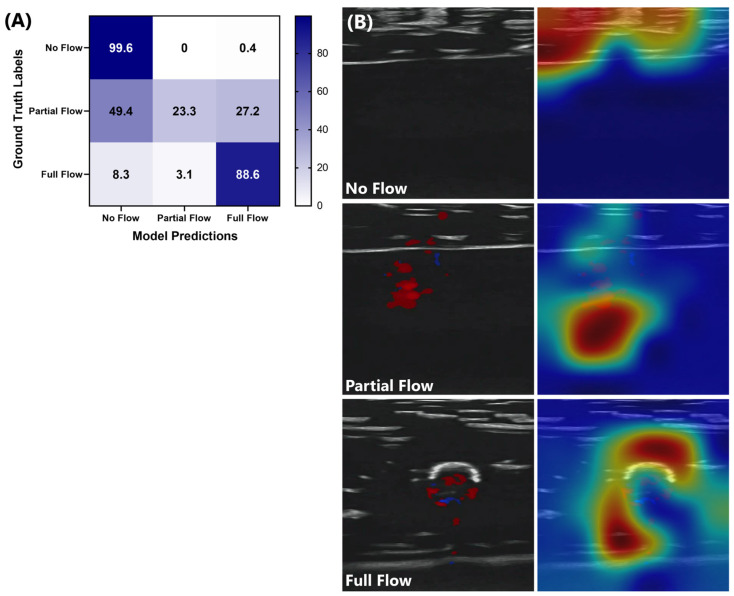
Three-category ShrapML performance for tracking junctional vessel occlusion. (**A**) confusion matrix for three categories—no flow, partial flow, and full flow—ShrapML models with affine transformations for data augmentation. (**B**) GradCAM overlays for representative ultrasound images for each of three categories. Areas with high relevance to model predictions are highlighted by red-yellow overlays, while lower relevance regions are highlighted in blue-green.

**Figure 4 bioengineering-11-00109-f004:**
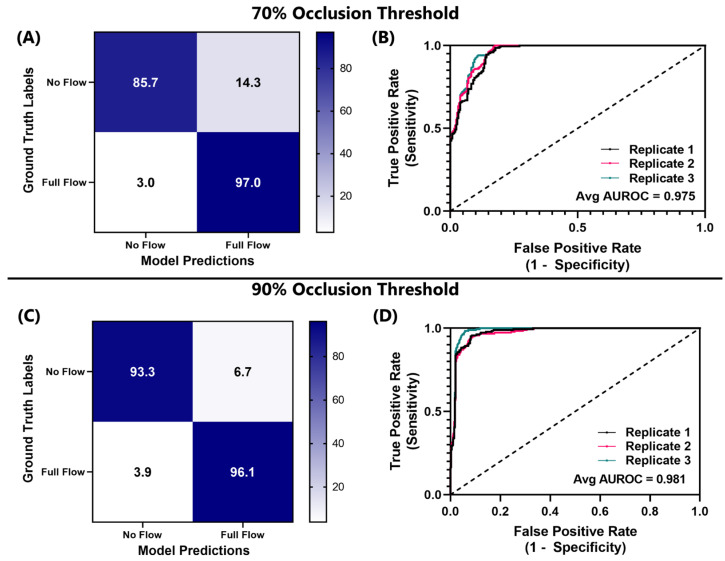
Confusion matrix and receiver operating characteristic (ROC) curve for ShrapML models tracking junctional vessel occlusion in swine image sets. Results are shown for three replicate trained models, shown as average values for the (**A**,**C**) confusion matrix and (**B**,**D**) individual ROC curves for a (**A**,**B**) 70% or (**C**,**D**) 90% occlusion threshold.

**Figure 5 bioengineering-11-00109-f005:**
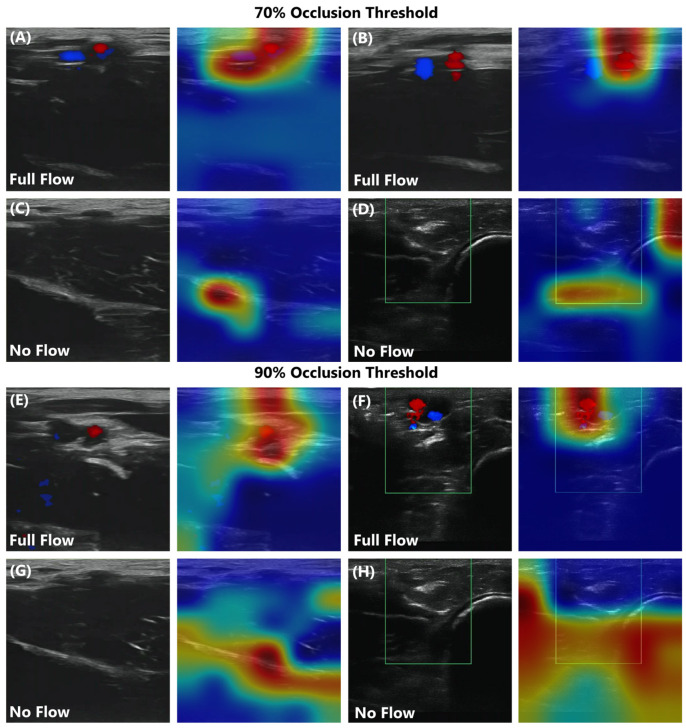
GradCAM overlays for ShrapML models trained with swine image sets. Representative images are shown for (**A**–**D**) a 70% or (**E**–**H**) a 90% occlusion threshold. Full-flow and no-flow ultrasound representative image categories are shown without and with GradCAM overlays. Areas with high relevance to model predictions are highlighted by red-yellow overlays, while lower relevance regions are highlighted in blue-green. Color Doppler overlay region is shown as a green box for images captured with Sonosite Edge.

**Table 1 bioengineering-11-00109-t001:** Performance metric summary for ShrapML models with training sets split at different pressure reduction thresholds. Models consisted of two categories—full flow and no flow—with affine transformations randomly applied for data augmentation. Metrics are shown as average results for *n* = 3 trained models. Color map is overlayed on each row to highlight the higher performance metrics from minimum (no color) to maximum (green).

	Distal Pressure Reduction Percent Threshold from Full Flow				
	50%	60%	70%	80%	90%
Accuracy	0.937	0.924	0.935	0.885	0.904
Precision	0.940	0.889	0.897	0.833	0.861
Recall	0.958	0.996	0.994	0.997	0.978
Specificity	0.905	0.823	0.864	0.744	0.821
F1 Score	0.948	0.939	0.943	0.907	0.916

**Table 2 bioengineering-11-00109-t002:** Performance metrics for MobileNetV2 and ShrapML models for ultrasound tracking of junctional vessel occlusion. Average (n = 3 trained models) performance metrics and standard deviations are shown for MobileNetV2 and ShrapML models without and with data augmentation.

	MobileNetV2				ShrapML			
	Without Data Augmentation		With Data Augmentation		Without Data Augmentation		With Data Augmentation	
	Average	Standard Deviation	Average	Standard Deviation	Average	Standard Deviation	Average	Standard Deviation
Accuracy	0.853	0.072	0.845	0.117	0.759	0.172	0.934	0.010
Precision	0.976	0.042	0.993	0.006	0.718	0.197	0.986	0.012
Recall	0.675	0.144	0.646	0.275	0.859	0.017	0.859	0.032
Specificity	0.990	0.018	0.996	0.004	0.683	0.316	0.991	0.008
F1 Score	0.794	0.114	0.757	0.223	0.771	0.120	0.918	0.014

**Table 3 bioengineering-11-00109-t003:** Summary of performance metrics for the ShrapML trained with swine image sets for tracking junctional vessel occlusion. Results are shown as average and standard deviations across three replicate trained models for a 70% or 90% occlusion threshold.

	70% Occlusion Threshold		90% Occlusion Threshold	
	Average	Standard Deviation	Average	Standard Deviation
Accuracy	0.909	0.015	0.942	0.029
Precision	0.971	0.025	0.980	0.014
Recall	0.857	0.040	0.933	0.032
Specificity	0.970	0.027	0.961	0.027
F1 Score	0.910	0.017	0.956	0.023

## Data Availability

The datasets presented in this article are not readily available because they have been collected and maintained in a government-controlled database that is located at the US Army Institute of Surgical Research. As such, these data can be made available through the development of a Cooperative Research and Development Agreement (CRADA) with the corresponding author. Requests to access the datasets should be directed to Eric Snider, eric.j.snider3.civ@health.mil.

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
