# Peer review of "Toward Smart, Automated Junctional Tourniquets—AI Models to Interpret Vessel Occlusion at Physiological Pressure Points"

_bioengineering, 2024, doi:10.3390/bioengineering11020109_

Round 1

Reviewer 1 Report

Comments and Suggestions for Authors

The paper’s subject is interesting and relevant. In the paper, the classification algorithm for the definition site to hemorrhage for its prevention and stopping. This algorithm is developed based on a neural network. The neural network was developed based on images captured from a custom-tissue mimicking phantom and an ex-vivo swine model of the inguinal region as pressure is applied using an ultrasound probe with and without color doppler overlays.

The abstract reflects the paper's context. The result is presented and discussed clearly. It is evaluated well. The introduction can be improved by citation and discussion of the studies of trends in applying AI-based methods in medicine. In particular, the correlation of the presented study with precision medicine, and smart medicine. For example, some publications in the journal Bioengineering can be considered (https://www.mdpi.com/2306-5354/10/7/838, https://www.mdpi.com/2306-5354/10/7/823 ). The achieved result has acceptable accuracy. The algorithm can be described in more detail. It is very well, that the authors consider the limitations of the proposed algorithm in the discussion’s section.

Author Response

Reviewer 1

  1. The abstract reflects the paper's context. The result is presented and discussed clearly. It is evaluated well. The introduction can be improved by citation and discussion of the studies of trends in applying AI-based methods in medicine. In particular, the correlation of the presented study with precision medicine, and smart medicine. For example, some publications in the journal Bioengineering can be considered (https://www.mdpi.com/2306-5354/10/7/838, https://www.mdpi.com/2306-5354/10/7/823 ). The achieved result has acceptable accuracy. The algorithm can be described in more detail. It is very well, that the authors consider the limitations of the proposed algorithm in the discussion’s section.

We appreciate the reviewer evaluating this manuscript for publication. We have added further literature review on how this effort fits within the greater body of work in the AI-precision medicine space. This can be found in the introduction starting at line 78. Based on reviewer recommendation, we have also added more description of the AI architecture starting on line 191 in the methods section.

Reviewer 2 Report

Comments and Suggestions for Authors

1. The work demonstrates a significant application of artificial intelligence for monitoring junctional vessel occlusion. However, there are a few areas where the authors could improve the article to make it more effective.

2. Consider adding a concluding sentence to summarize the key findings (quantitative) at the end of the Abstract section.

3. Write structure of paper (sections) at the end of Introduction section.

4. Specify the rationale behind selecting a swine model and explain the relevance of the chosen anatomical region (lumbar area to knees).

5. Specify the rationale behind selecting the ShrapML model and MobileNetV2 for this study.

6. Consider providing a brief explanation of the choice of the RMSProp optimizer and its parameters.

7. Provide a brief explanation of why GradCAM overlays are used and their role in model evaluation.

8. The reliance on phantom data raises concern regarding the model's real-world applicability. Provide a justification for using phantom data, emphasizing the reliability of the model.

Comments on the Quality of English Language

Moderate editing of English language required

Author Response

  1. The work demonstrates a significant application of artificial intelligence for monitoring junctional vessel occlusion. However, there are a few areas where the authors could improve the article to make it more effective.

We appreciate the reviewer evaluating this manuscript for publication and have addressed each comment below.

  1. Consider adding a concluding sentence to summarize the key findings (quantitative) at the end of the Abstract section.

We have modified the abstract on line 24 to better indicate the conclusions and quantitative metrics for the research performed in this manuscript.

  1. Write structure of paper (sections) at the end of Introduction section.

We have outlined at the end of the introduction at line 101 the experimental structure taken in this paper to better indicate the section organization of the paper.

  1. Specify the rationale behind selecting a swine model and explain the relevance of the chosen anatomical region (lumbar area to knees).

The region was selected as it provided sufficient area for experimental setup and swine were selected due to their cardiovascular similarity to human. Clarification on these comments were addressed in the methods section starting at line 144.

  1. Specify the rationale behind selecting the ShrapML model and MobileNetV2 for this study.

Additional details explaining more of the rational for model selection were added to methods section 2.5 starting at line 187.

  1. Consider providing a brief explanation of the choice of the RMSProp optimizer and its parameters.

The RMSprop optimizer was chosen by Bayesian optimization for the custom classification network in our previous work, which also determined the 0.001 learning rate and total of 100 training epochs. To avoid overfitting, we have added  validation patience as a training option which allows the model to stop training before completing the 100 epochs. Details on the RMSprop selection were added at line 194 in the methods.  

  1. Provide a brief explanation of why GradCAM overlays are used and their role in model evaluation.

The reason for using GradCAM overlays and how they can support model explainability are on lines 225 in the methods. We have added examples to how other research studies have used GradCAM overlays in the same section.

  1. The reliance on phantom data raises concern regarding the model's real-world applicability. Provide a justification for using phantom data, emphasizing the reliability of the model.

We are aware of the limitations encountered using a phantom model, this is discussed starting at line 395 in the discussion section. For the current study, we also tested our vessel occlusion prediction models on an ex-vivo- swine model. We have also  previously shown how supplementing training from animal subjects with phantom images maintains high accuracy predictions while requiring less images from the live subject.